# A novel Fish Protein Hydrolysate supplement enhances honey bee foraging activity and colony strength: A pilot study

**Hatem Sharaf El-Din[1], Hossam Radawan[2], Hamed A. Ghramh[3], Yahya Al Naggar [3]***

**1** Department of Economic Entomology and Pesticides, Faculty of Agriculture, Cairo University, Giza, Egypt, **2** Bee Venom Production Association, Kafr El-Zayat, Al Gharbiyah, Egypt, **3** Center of Bee Research and its Products, King Khalid University, Abha, Saudi Arabia

* yehia.elnagar@science.tanta.edu.eg

## Abstract

Honey bees require diverse nectar and pollen sources for optimal nutrition, especially during late winter and early spring. This study evaluated the effect of a tuna fish protein hydrolysate (FPH)-based dietary supplement on honey bee colonies during winter season. FPH was extracted with Spasmodigestin® tablets, which increased protein content and the availability of 15 amino acids. Colonies were fed 2% FPH extract in sugar syrup (50% W/V) weekly for six weeks, while control colonies received only sugar syrup. The size of brood, honey, and bee bread areas were measured every two weeks, while foraging activity (number of incoming and outgoing foragers) was evaluated weekly. FPH-fed colonies showed improved foraging activity (130 ± 10.65%), larger brood areas (116 ± 11.10%), and higher reserves of honey (152 ± 61.87%) and bee bread (132 ± 61.39%). These findings highlight the potential of tuna FPH-based supplements to strengthen honey bee colonies during periods of limited forage availability.

## 1. Introduction

Honey bees (*Apis mellifera*) play a crucial role in maintaining the balance of ecosystems and supporting global food production. As key pollinators, they facilitate the reproduction of flowering plants by transferring pollen from one flower to another, enabling the formation of fruits and seeds [1,2]. In addition to their role in agriculture, honey bees contribute to biodiversity by supporting wild plant populations, which provide habitat and food for many other species. Thus, the preservation of honey bee populations is vital for ecological health, food security, and biodiversity [3,4].

Nutrition for honey bees primarily comes from pollen (providing protein, fats, vitamins, and minerals) and nectar (main carbohydrate source) [5,6]. Nectar also contains lipids, organic acids, minerals, and proteins, but in low concentrations [7,8]. Poor nutrition, on the other hand, is one of the many factors associated with honey

**Data availability statement:** ll relevant data are within the manuscript and its Supporting Information files.

**Funding:** This work was funded by the Deanship of Research and Graduate Studies at King Khalid University through the Large Research Project number (RGP2/540/45) awarded to Yahya Al Naggar.

**Competing interests:** The authors have declared that no competing interests exist

bee colony losses because a lack of proper nutrition such as a lack of availability or diversity of floral resources, exacerbates stress caused by other factors such as pesticides and diseases, increasing the chance of colony collapse [9–11]. For example, a lack of diversity in diet can weaken bees' immune systems [12], leaving them more susceptible to diseases and parasites [13–15]. Furthermore, poor nutrition causes bees to have shorter lifespans and reduced foraging activity, which has an impact on the colony's general health [16–18]. As a result, ensuring bees have access to a range of high-quality floral resources is key to their survival and maintaining the vital ecological services they perform [19]. However, this may not be feasible during the winter and early spring.

During winter, honey bees rely on stored resources for survival, but sometimes these stores may not be sufficient, particularly in regions with long, harsh winters or where fall foraging is limited. To compensate for nutritional deficiencies, certain commercial dietary supplements made up of either real pollen or patty-shaped pollen substitutes (beer yeasts, soybean meals) have been developed [20,21]. Beekeepers feed their colonies these dietary supplements to encourage brood raising in the late winter or early spring and to ease dietary stress [21–23]. However, it's important to choose high-quality supplements that closely mimic the nutritional value of natural pollen to ensure the best outcomes for the bees [24]. Beekeepers currently use a wide range of feeding techniques and diets. They typically deposit pollen replacements in patty form close above the brood nest [6,25]. They may also use dry powder feeders outside the hives [23].

Pollen substitutes must be inexpensive, long-lasting, and easily to feed colonies to comply with modern beekeeping practices. However, a good pollen substitute should be palatable and beneficial to bees' health [23]. The most cost-effective strategy is not always the best for bee health. Natural pollen, for example, is difficult to obtain in considerable amount at a low cost and involves the danger of disease transmission, despite being suitable for pollen substitutes [26,27]. Unfortunately, no substitute diet has been developed that can replace the nourishment provided by pollen [27,28]. Beekeepers spend a lot of time and supplies controlling pollen deficits in their colonies. Despite this, no extensive body of research or literature uniformly supports their use. It is worth noting that most available and commercial pollen substitute components, including soybean flour, brewer's yeast, chickpea, maize, sorghum, and wheat flour, are botanically derived and consumed by humans, as are other animal-derived ingredients such skim milk and egg yolk powder. Furthermore, pollen grains are now employed in bread production and ingested by humans [29], as is table sugar, which is the primary supply of carbohydrates during nectar shortages. So, bees regularly consume human food. As a result, it is crucial to discover a new alternative animal/fish protein source that is widely available, affordable, and simple to utilize. Studies have shown that insect protein hydrolysates, can serve as sustainable protein sources for various animals, due to their high digestibility and rich amino acid profile [30–34]. However, its potential application as a food supplement for honey bee colonies has not yet been examined.

Fish protein hydrolysate (FPH) is a substance formed by the enzymatic breakdown of fish proteins into smaller peptides and amino acids. It is typically manufactured from fish processing leftovers such as heads, bones, and skin, with the proteins hydrolyzed using enzymes [35,36]. This technique yields a very nutritious product that can be utilized in a variety of applications [37]. FPH comes in liquid or powder form and has between 81–93% protein, 3–8% ash, less than 5% fat, and 1–8% humidity [38]. It is commonly used as a supplement in animal feeds, especially for aquaculture [39], poultry [40], and livestock [41], because it promotes development and immunological function. FPH is also utilized as a biostimulant in agriculture to increase plant growth and soil health [42]. Given that protein and amino acids are critical for honey bee growth, development, immunity, and general colony health, bees require ten essential amino acids (e.g., valine, leucine, lysine) for muscle development, enzyme production, and immune responses [43,44]. As a result, FPH could be a vital source of nutrition for honey bees, particularly during periods of limits pollen availability. However, its efficacy is contingent on characteristics such as palatability and quality. More research is needed to optimize its application and comprehend the long-term consequences on bee health.

There has only been one study to date that assessed the efficacy of a nutritional supplement in nucleus colonies that consisting of only 5 combs [45]. This nutritional supplement is made up of two significant phytochemicals, *p*-coumaric acid (CUM) and abscisic acid (ABA), together with FPH and omega fatty acids and applied with sugar syrup. The data obtained demonstrate that nutritional supplement is appealing and non-harmful to bees and greatly increases the amount of open brood and pollen reserves of bee nuclei compared to nuclei fed sugar syrup only. On the other hand, there are no noticeable variations in honey reserves and bee protein content [45]. However, this study was conducted during the summer months of December 2018 and January 2019, in Argentina. As a result, field trials during the winter season, when floral resources are few and foraging opportunities are limited, are required to support the findings of this study. It is also important to examine the potential consequences of foraging actions.

No studies have specifically evaluated the effectiveness of an FPH-based supplement on honey bee colonies during winter, nor have they examined its impact on foraging activity. This study aims to address this gap by assessing the efficacy of a cost-effective tuna FPH extract as a dietary supplement for honey bees. We hypothesized that supplementing winter colonies with tuna FPH extract would enhance foraging activity and improve colony strength, as measured by increased brood area and stored honey and bee bread reserves [46].

## 2. Materials and methods

### 2.1. Fish Protein Hydrolysates (FPH) extraction

To obtain the tuna FPH, commercial canned skipjack tuna (*Katsuwonus pelamis*) (130 grams) was utilized. The contents of each shredded tuna can were transferred to a glass jar and four tablets of Spasmodigestin® (Ingredients & Concentrations: Papain 100 mg, Sanzyme 3500–36 mg, Sodium dehydrocholate — 10 mg, Dicyclomine hydrochloride — 5 mg, Simethicone 30 mg) obtained from a local pharmacy and incubated at 38 °C for 12 hours according to [47] with modification (incubate for 12 hours at 38°C rather than 4 hours at 55°C). The extract was then filtered and stored at 4 ºC (Fig 1). To evaluate the efficacy of the extraction procedure, aliquots of the FPH extract were examined for total protein, lipid, and carbohydrate content, as well as amino and fatty acid components, in comparison to canned tuna. We used Spasmodigestin®, a well-balanced combination preparation with digestive, choleretic, antispasmodic, and anti-inflammatory effects. It feeds the body with papain, a proteolytic enzyme, as well as Sanzyme 3500, a multi-enzyme complex mostly composed of protease, amylase, lipase, and cellulase. These enzymes work on the food bolus in the digestive tract, catalyzing its breakdown into smaller and more digestible components. In addition, papain has been shown to alleviate lipid accumulation and inflammation in high-fat diets [48]. Furthermore, Spasmodigestin® is cost-effective and available in the local market. The protein content of FPH of canned tuna before and after extraction was determined using the Kjeldahl method [49] which estimate the protein content of a sample based on its nitrogen content. While the total fat content was assessed

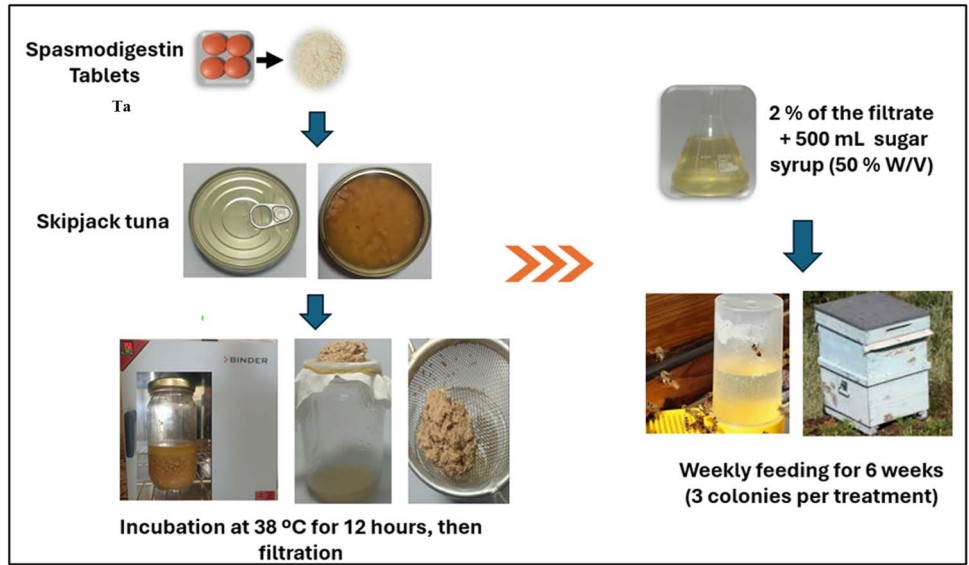

**Fig 1. Displays the diagrammatic steps of tuna fish protein hydrolysate (FPH) extraction and the experimental design, including the time of feeding and the number of colonies employed per treatment.**

using the Folch method [50]. Total Carbohydrates content was determined colorimetry by phenol sulfuric acid method [51] which is a colorimetric technique commonly used to quantify carbohydrates and polysaccharides.

Amino acids content was determined using HPLC (Agilent 1260 series) following [52–55]. In brief, 0.1 g of the sample was mixed with 5 mL $H_2O$ and 5 mL of Hydrochloric acid (HCl) (Note: final concentration. of HCl is 6 M) and then heated at 120°C for 24 hrs and then filtered. Finally, 1 mL of the filtrate was dried and resuspended in 0.1 M HCl and injected into HPLC. The separation was carried out using Eclipse Plus C18 column (4.6 mm x 250 mm i.d., 5 μm). The mobile phase consisted of buffer (sodium phosphate dibasic and sodium borate), pH 8.2 (A) and Acetonitrile (ACN): Methanol (MeOH): Water ($H_2O$) 45:45:10 (B) at a flow rate 1.5 ml/min.

While Fatty acid methyl esters (FAME) of FPH before and after extraction were produced by an alkali-catalyzed reaction between fats and methanol in the presence of 2M potassium hydroxide ([56] and injected in hexane to GC (model 7890B from Agilent Technologies). The GC model 7890B from Agilent Technologies was equipped with flame ionization detector at Central Laboratories Network, National Research Centre, and Cairo, Egypt. Separation was achieved using a Zebron ZB-FAME column (60 m x 0.25 mm internal diameter x 0.25 μm film thickness). Analyses were carried out using hydrogen as the carrier gas at a flow rate of 1.8 ml/min at a split-1:50 mode, injection volume of 1 μl and the following temperature program: 100 °C for 3 min; rising at 2.5 °C/min to 240 °C and held for 10 min. The injector and detector (FID) were held at 250 °C and 285 °C, respectively.

## 2.2. Honey bees

Colonies of *A. mellifera carnica* headed by mated sisters' queens were maintained in the apiary yard and were used in February 2024. Colonies had no visible honey bee diseases and each colony had 5 combs (containing 3 combs of brood + 2 combs of honey and bee bread). All colonies were treated with formic acid 60% in autumn (September) presented on carton paper for *Varroa* mites' control and *Varroa* levels have been checked before the beginning of the experiments.

## 2.3. Experimental design

It is noteworthy to mention that during the 1st preliminary trials of tuna FPH extraction indicated above, we carried out three-day incubation of the extraction process at 38 °C, then eight honey bee colonies were used. Each colony was fed with a half liter of sucrose solution (50% w/v), either alone for the control group or combined with the FPH extract at ratios of (1, 2 and 4%). Each treatment or control group consisted of two colonies. Initially, colonies were examined to determine whether honey bees could consume sugar syrup with varying FPH concentrations. Bees only consumed sucrose solution containing 1% and 2% FPH extract, However, consumption was considerably slower than in the control group. Then, we modified the feeding method by supplying these three concentrations with sugar candy, but bees did not accept all of them, possibly because it was unpalatable owing to the smell, thus no consumption occurred. After that, we modified the extraction process and carried out the incubation for only 12 h at 38 °C as mentioned in section 2.1. We noticed that the extract was diluted and smelled much better as the previous extract smelled like digested fish. Therefore, we designed a new experiment in which three colonies were fed sucrose syrup free of FPH extract (control) and three colonies were weekly fed sucrose solution supplemented with 2% of FPH extract for 6 weeks as illustrated in (Fig 1). We selected 2% of FPH rather than 1% for a potential better result. Biological measures were applied to all experimental hives, including areas with sealed brood, honey, and bee bread, before the start of the experiment and every two weeks for six weeks as recommend [57]. Each frame was photographed from both sides with a digital camera, and pictures were imported into the ImageJ (version 1.54 f) software program to measure and calculate the area of the comb ($cm^2$) of honey, brood, and beebread. Foraging activity was also evaluated weekly throughout the six-week assessment, with each colony monitored for two minutes at 12 pm and the number of foragers leaving and returning recorded.

## 2.4. Statistical analysis

All statistical analyses and data visualization were performed using R studio version 4.4.1 [58]. To compare the change in the comb areas of brood, honey, and bee bread as well as foraging activity (number of incoming and outgoing foragers) between treated and non-treated colonies and before and after treatment, we used ANOVA (Type II) tests in a linear mixed model (LMM). Treatment, time of assessment were used as independent, fixed factors (predictors) while the colony was included as a random factor. To test for significant interactive effects of treatment, time of assessment of areas of brood, honey, and bee bread, and foraging activity, we inspected the treatment × time interaction terms in all models.

## 3. Results

### 3.1. Total protein, fats, and carbohydrates content of tuna FPH extract

Extraction of tuna FPH using Spasmodigestin® tablets led to a 2.76-fold increase in total protein content (4.75 to 13.11%), while it led to a 1.44 (56.9 to 39.3%) and 1.18 (2.09 to 1.77 g/100 g)-fold decrease in total fats and carbohydrates content, respectively (Table 1).

### 3.2. Amino and fatty acid content of tuna FPH extract

Quantitative analysis utilizing HPLC revealed the presence of 16 amino acids in tuna FPH. Almost all amino acids increased when tuna FPH was extracted using Spasmodigestin® tablets (Table 2) For example, concentration of aspartic

**Table 1. Total Protein, total fats, and carbohydrates content in tuna Fish protein hydrolysate (FPH) before and after extraction.**

|  | Total protein (%) | Total fats (%) | Carbohydrates (g/100g) |
|---|---|---|---|
| **Before** | 4.75 | 56.9 | 2.09 |
| **After extraction** | 13.11 | 39.3 | 1.77 |

acid, glutamic acid, serine, arginine, and leucine increased by 1.83, 3.97, 14.27, and 12.81-fold, respectively. While the concentrations of histidine and valine amino acids decreased by 1.46 and 2.16-fold. Cystine amino acid was not detected before or after extraction, whereas methionine amino acid was found in an increased concentration only after extraction (Table 2).

A semi-quantitative analysis of fatty acids using gas chromatography revealed the presence of 16 fatty acids in tuna FPH. Extraction with Spasmodigestin® tablets indicated minimal variations in the levels of fatty acids, while some increased such as palmitic acid, stearic acid, and docosahexaenoic acid (DHA), decreased such as myristic acid and linoleic acid, or remained unchanged such as palmitoleic acid, margaric acid, and cis-11-eicosenoic acid (Table 2).

### 3.3. Effect of tuna FPH extract on colony strength and foragers activity

Weekly feeding of honey bee colonies during the winter season (February-March) on a dietary supplement based on tuna FPH extract for six weeks revealed a significant increase in the comb area of stored honey (152 ± 61.87%), which also significantly increased with time, but there was no significant interaction term. There was a significant interaction term (treatment x time) for both the comb area of bee bread and bee brood ($p < 0.05$: see Table 3 for complete statistical details), indicating that the increase in brood (116 ± 11.10%) and bee bread (132 ± 61.39%) areas in response to tuna FPH extract feeding was time-dependent (Fig 2, S1 Table in S1 File). We checked the colonies after the termination of the treatment to ensure the quality and safety of honey and there was no odour of this FPH. Moreover, we tested some stored honey, and everything was similar to control colonies. Also, there have been no discernible changes in bee behavior.

Feeding had a noticeable effect on forager activity as well. There was an increase in the number of entering (130 ± 14.91%) and exiting foragers (126 ± 10.89%), although this rise was only significant for incoming foragers ($p < 0.05$:

**Table 2. Amino acids and fatty acids content in tuna fish protein hydrolysate (FPH) before and after extraction.**

| Amino acid | Conc. (µg/ml) | | Fatty acid | RT | Area Sum % | |
|---|---|---|---|---|---|---|
| | Before | After | | Before | After | |
| Aspartic acid | 46.50 | 85.19 | Myristic acid | 21.192 | 0.06 | 0.02 |
| Glutamic acid | 106.76 | 424.89 | Palmitic acid | 26.83 | 10.76 | 11.27 |
| Serine | 40.77 | 169.32 | Palmitoleic acid | 28.018 | 0.08 | 0.08 |
| Histidine | 4604.26 | 3139.42 | Margaric acid | 29.562 | 0.09 | 0.09 |
| Glycine | 70.51 | 154.29 | cis-10-Heptadecenoic acid | 30.565 | 0.05 | 0.05 |
| Threonine | 31.25 | 110.07 | Stearic acid | 32.308 | 3.69 | 4.06 |
| Arginine | 50.34 | 718.49 | Oleic acid | 33.166 | 26.43 | 26.93 |
| Alanine | 173.48 | 588.09 | Linoleic acid | 34.951 | 52.25 | 50.5 |
| Tyrosine | 37.69 | 210.42 | Linolenic acid | 36.937 | 5.33 | 5.24 |
| Cystine | 0.00 | 0.00 | Arachidic acid | 37.309 | 0.34 | 0.35 |
| Valine | 805.25 | 371.59 | *cis*-11-Eicosenoic acid | 38.017 | 0.22 | 0.22 |
| Methionine | 0.00 | 392.95 | Behenic acid | 42.087 | 0.41 | 0.48 |
| Phenylalanine | 46.21 | 372.79 | Eicosapentaenoic acid (EPA) | 43.594 | 0.01 | 0.08 |
| Isoleucine | 32.88 | 204.22 | Tricosanoic acid | 44.377 | 0.04 | 0.05 |
| Leucine | 70.61 | 904.65 | Lignoceric acid | 46.602 | 0.14 | 0.13 |
| Lysine | 128.47 | 338.76 | Docosahexaenoic acid (DHA) | 48.717 | 0.09 | 0.44 |
| Proline | 62.90 | 112.76 | | | | |

RT. Retention time

**Table 3. Results from LMM analyses testing effects of winter weekly feeding of honey bee colonies (n = 3) for 6 weeks on sugar syrup spiked with 2% fish protein hydrolysate (FPH) extract on comb areas (cm²) of honey, bee bread, brood as well as on foraging activity (number of ongoing and outgoing foragers) at different time points. In bold are treatment or/and time effects that were significantly different from control (p < 0.05).**

|  |  | (Type II Wald chi-square tests) (P-value) |  |  |  |  |
| --- | --- | --- | --- | --- | --- | --- |
| Source | Df | Honey comb area | Beebread com area | Brood comb area | No. of incoming foragers | No. of outgoing foragers |
| **Treatment** | 1 | **<0.001** | **0.004** | **<0.001** | **0.04** | 0.05 |
| **Time** | 1 | **<0.001** | **<0.001** | **<0.001** | **<0.001** | **<0.001** |
| **Treatment x time** | 1 | 0.05 | **<0.001** | **<0.001** | 0.06 | 0.05 |

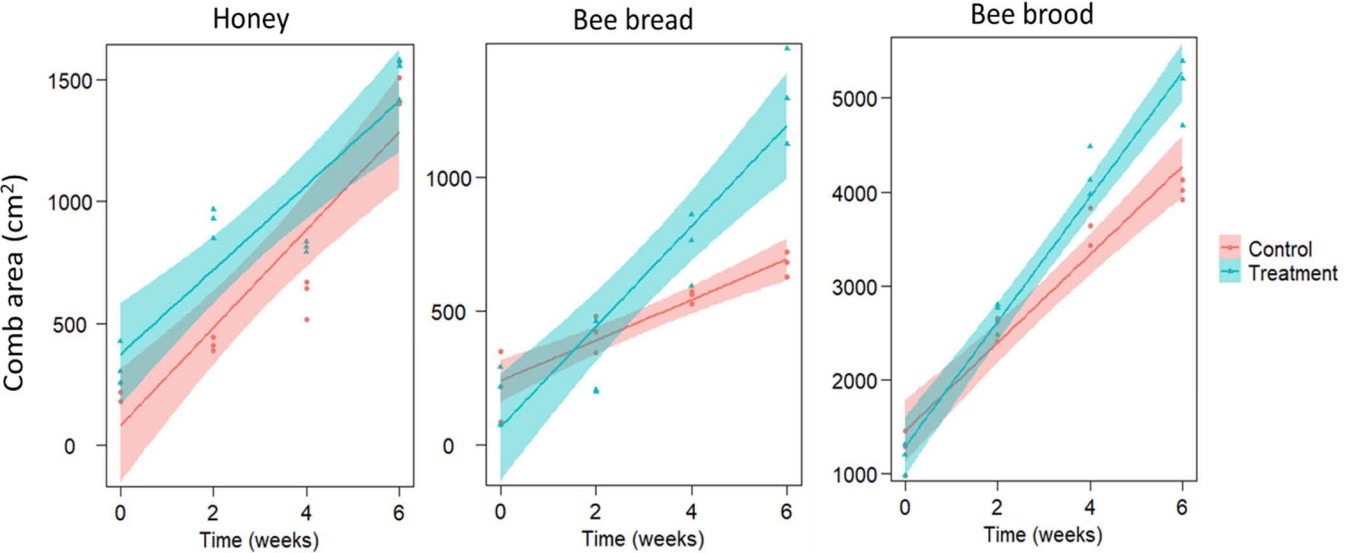

**Fig 2. Effects of winter weekly feeding of honey bee colonies (n = 3) for 6 weeks on sugar syrup spiked with 2% fish protein hydrolysate (FPH) extract on comb areas (cm²) of honey (left), bee bread (middle), and bee brood (right) before and after treatment.** Smoothed lines show the predicted relationships of the LMMs, and shaded areas indicate the 95% confidence intervals. Dots and triangles show raw data. Comb areas of honey, bee bread, and bee brood significantly increased (p < 0.05) in treated colonies compared to control. For statistical details (see Table 3).

see Table 3 for complete statistical details), indicating an improvement in foraging activity that will affect colony strength. It's also worth noting that forager activity increased dramatically over time, confirming the typical increase in forager activity with weather improvements (Fig 3, S2 Table in S1 File).

## 4. Discussion

Fish are commonly recognized as a rich and nutrient-dense food. When compared to other foods derived from animals, such meat, poultry, and eggs, fish is low in cholesterol and saturated fats. It has been suggested that fish and fish products are a crucial component of a balanced diet, particularly when they take the place of other meals high in cholesterol and saturated fats but low in protein [59]. Several attempts have been made recently to produce commercially valuable food ingredients from the protein-rich fish industry by-product wastes and underutilized fish proteins [35]. The nutritional composition, amino acid profile, and antioxidant properties of FPH have attracted the interest of food experts. Because FPHs include vital minerals and bioactive elements, they are used in a variety of industrial applications [60].

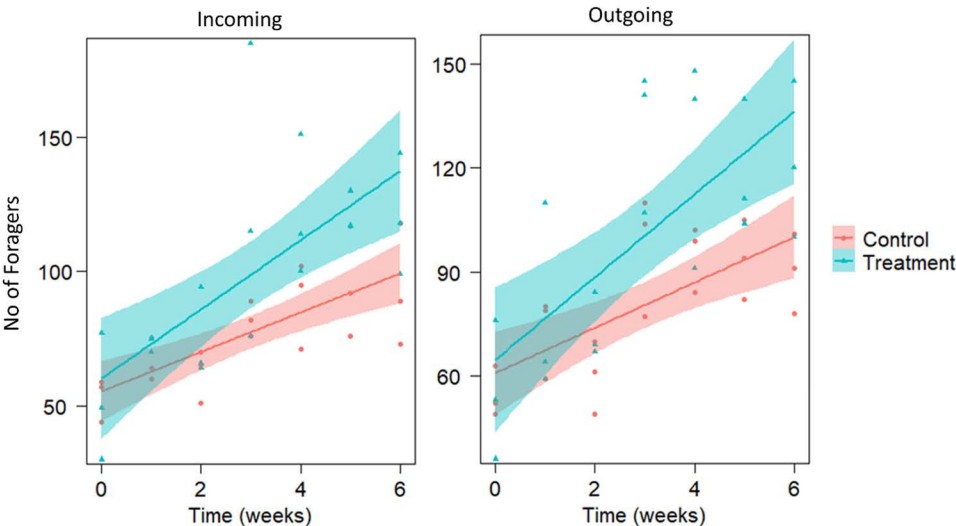

**Fig 3. Effects of winter weekly feeding of honey bee colonies (n = 3) for 6 weeks on sugar syrup spiked with 2% tuna fish protein hydrolysate (FPH) extract on foragers activity: number of incoming foragers (left) and number of outgoing foragers (right) before and after treatment.** Smoothed lines show the predicted relationships of the LMMs, and shaded areas indicate the 95% confidence intervals. Dots and triangles show raw data. Number of incoming foragers significantly increased (p < 0.05) in treated colonies compared to control. For statistical details (see Table 3).

In the current study, the total amount of protein and concentrations of 15 amino acids increased in tuna FPH following extraction with Spasmodigestin° tablets, while total lipids and carbohydrates decreased. The current finding is consistent with previous research, which has shown that papain, an ingredient of Spasmodigestin° tablets, plays a significant role in protein digestion and alleviates lipid accumulation and inflammation in high-fat diets [61]. Of the twenty naturally occurring amino acids, at least ten must be present in an insect's diet. These ten, called essential amino acids, include lysine, tryptophan, histidine, phenylalanine, leucine, isoleucine, threonine, methionine, valine, and arginine [62,63]. The current study revealed that extracting tuna FPH using Spasmodigestin° tablets enhanced the levels of seven essential amino acids: lysine, phenylalanine, leucine, isoleucine, threonine, methionine, and arginine, confirming the effectiveness of the extraction method used.

Honey bees consume pollen protein, which provides critical amino acids for the synthesis of crucial immunological compounds such antimicrobial peptides (AMPs), which are needed in immune pathways [64,65]. Consuming nectar or honey also supplies energy for metabolic activities necessary for innate humoral and cellular immune responses [66]. However, pollen substitutes play a crucial role in beekeeping, particularly during periods of pollen scarcity. The effects of high-protein feeds on honey bee growth, development, performance, and overwintering using eight protein sources (10% skim milk powder, 30% honey and 60% from one of the protein ingredients including lentil flour, soybean flour, soybean meal, wheat gluten, and fish meal) were investigated [67]. Pollen had the highest consumption, while fish meal had the lowest. Colonies fed pollen had the highest egg-laying area (22,636 cm²) and honey production, whereas fish meal resulted in the lowest values (13,052 cm²). In the current study, providing 2% of tuna FPH extract to honey bee colonies with sucrose solution in late winter increased their strength by expanding their brood area (116 ± 11.10%), storing more honey (152 ± 61.87%) and bee bread (132 ± 61.39%), and stimulating their foraging activity (130 ± 10.65%). These results are not compatible with [67], but they are consistent with a recent study in which bees were fed a supplement diet comprising *p*-coumaric and abscisic acid, FPH, and omega fatty acids in nucleus colonies. [45]. Interestingly, the present study differs from the previous studies in that it was conducted

during the winter season rather than the summer, and only FPH was employed, demonstrating the efficacy of this novel dietary supplement based on tuna FPH extract for honey bee health. These positive outcomes and the disparity with [67] could be attributed to the way bees get the protein supplement. Because previous studies revealed differences in absorption rates between liquid and solid protein supplements. Liquids deliver a more concentrated protein dosage. They are often minimal in calories and can be quickly eaten and absorbed without the need to chew or digest more complex ingredients [68,69]. Future studies ought to validate this assumption by comparing the efficacy of this tuna FPH solution to a typical pollen substitute patty.

The increase in brood area observed in the current study after feeding on sucrose solution containing 2% tuna FPH extract could be attributed to the diet's high protein content and ease of digestion, as proteolytic enzymes reduce peptide size, making hydrolysates the most available amino acid source for various physiological functions [35]. Furthermore, the extraction of tuna FPH resulted in a drop in total fats and levels of certain fatty acids, and prior research demonstrated that honey bees prefer low-fat diets over high-fat diets [70]. Furthermore, bumble bees have been found to clearly distinguish between regular pollen and pollen with higher fat content, and they did exhibit a clear preference for normal pollen. Increased fat levels in the diet also negatively influenced their survival and reproduction [71].

Increased foraging behavior directly results from good nutrition at the colony level, which enhances the hive's general health and production [72,73]. For example, honey bees with improved protein nutrition, through supplemental feeding of protein-rich diets, showed a significant increase in foraging behavior [74]. Similarly, colonies that fed syrup supplemented with 2% of tuna FPH extract showed an increase in foragers activity that reflected on the amount of stored honey, bee brood, and bee bread and over all colony health. As strong bee colonies rear more brood and produce more honey than weak colonies. A significant positive correlation was established between the strength of the bee colony and brood amount, honey production, and between brood amount and honey production [46,75]. This suggests that protein availability plays a key role in regulating the energy levels and motivation of forager bees.

## 5. Conclusion

The extraction of tuna FPH using Spasmodigestin® tablets was highly effective since it enhanced the protein content and levels of approximately 15 amino acids while decreasing the fat content. Then, weekly feeding of honey bee colonies during the winter with sucrose solution containing 2% tuna FPH over 6 weeks resulted in an increase in forager activity, which translated into an increase in brood area as well as honey and bee bread reserves. Our findings shed light for the first time on the promising positive impact of a novel food supplement based on tuna FPH on honey bee health and colony strength. It is important to note that this study represent a pilot study with only six colonies and feeding for six weeks, and future studies should be undertaken on a wider scale and for longer periods. In addition, the effect of this unique diet on individual honey bee health (development of hypopharyngeal glands, fat body, body weight, and immunocompetence against various diseases and parasites) remains to be studied to gain a better understanding of its nutritional value when compared to regular pollen substitutes.

## Supporting information

**S1 File. The study generated raw data and mean±SEM (Table S1 & S2) for all assessed variables.**
(XLSX)

## Acknowledgments

The authors would like to thank Omnia Arafat Abdel Latif, Donia Hassan Ali, Rehab Ahmed Khair, and Nourhan Mohamed Khalil for helping in data collection. Authors also appreciated the suggestions raised by Dr. Kai Wang.

## Author contributions

**Conceptualization:** Hatem Sharaf El-Din, Hossam Radawan, Yahya Al Naggar.

**Data curation:** Hatem Sharaf El-Din.

**Investigation:** Hatem Sharaf El-Din, Hossam Radawan, Yahya Al Naggar.

**Methodology:** Hatem Sharaf El-Din, Hossam Radawan, Yahya Al Naggar.

**Project administration:** Hamed A. Ghramh.

**Software:** Yahya Al Naggar.

**Supervision:** Hamed A. Ghramh.

**Validation:** Hamed A. Ghramh.

**Visualization:** Yahya Al Naggar.

**Writing – original draft:** Hatem Sharaf El-Din, Yahya Al Naggar.

**Writing – review & editing:** Hatem Sharaf El-Din, Hamed A. Ghramh, Yahya Al Naggar.

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
