## [Decision Letter · Decision Letter 0]

21 Mar 2025

PONE-D-25-06788

A novel Dietary Supplement Based on Fish Protein Hydrolysate Extract Promotes Honey Bee Foraging Activity and Colony Strength: An applied pilot study

PLOS ONE

Dear Dr. Al Naggar,

Thank you for submitting your manuscript to PLOS ONE. After careful consideration, we feel that it has merit but does not fully meet PLOS ONE’s publication criteria as it currently stands. Therefore, we invite you to submit a revised version of the manuscript that addresses the points raised during the review process.

We look forward to receiving your revised manuscript.

Kind regards,

Muhammad Imran

Academic Editor

PLOS ONE

Journal Requirements:

2**.** Please include a separate caption for each figure in your manuscript.

Additional Editor Comments:

The study explores an interesting and novel approach to enhancing honey bee nutrition through a tuna-based fish protein hydrolysate (FPH) supplement, providing valuable insights into its effects on foraging activity and colony strength. However, several areas require significant revisions to improve the manuscript’s clarity and scientific depth. The introduction should provide a more detailed discussion of the nutritional components of FPH and the functional roles of amino acids in bee health. The methodology section needs further elaboration on the FPH extraction process, including the rationale for using Spasmodigestin® tablets, a clearer description of sugar syrup composition, and a quantification method for syrup consumption. The discussion should address the long-term effects of increased foraging activity on colony health and provide a comparison with existing commercial supplements. Additionally, addressing the potential environmental implications of utilizing fish by-products as a bee supplement would strengthen the manuscript. These revisions will significantly improve the manuscript’s clarity, impact, and relevance.

Reviewers' comments:

Reviewer's Responses to Questions

**Comments to the Author**

1. Is the manuscript technically sound, and do the data support the conclusions?

Reviewer #1: Partly

Reviewer #2: Yes

2. Has the statistical analysis been performed appropriately and rigorously? 

Reviewer #1: Yes

Reviewer #2: Yes

3. Have the authors made all data underlying the findings in their manuscript fully available?

Reviewer #1: Yes

Reviewer #2: No

4. Is the manuscript presented in an intelligible fashion and written in standard English?

Reviewer #1: Yes

Reviewer #2: Yes

5. Review Comments to the Author

Reviewer #1: The title is clear and informative, indicating the study's main focus and experimental outcome. However, it could be improved by making it more concise. Suggested revision:

"A Novel Fish Protein Hydrolysate Supplement Enhances Honey Bee Foraging Activity and Colony Strength: A Pilot Study.

The abstract is generally well-structured and includes the background, objective, methods, key findings, and conclusion. However, it could be improved by:

Briefly mention the duration of the feeding trial and the criteria for evaluating colony health.

Include specific quantitative outcomes to strengthen the findings (e.g., percentage increase in brood area or forager activity).

Use consistent terminology for "sugar syrup" (e.g., 50% W/V or 1:1 sugar:water ratio).

abstract revision "Honey bees require diverse nectar and pollen sources for optimal nutrition, especially during late winter and early spring. This study evaluated the effect of a tuna fish protein hydrolysate (FPH)-based dietary supplement on honey bee colonies. FPH was extracted using Spasmodigestin® tablets, increasing protein content and amino acid availability. Colonies were fed 2% FPH extract in sugar syrup weekly for six weeks, while control colonies received only sugar syrup. FPH-fed colonies showed improved foraging activity, larger brood areas, and higher reserves of honey and bee bread. These findings highlight the potential of tuna FPH-based supplements to strengthen honey bee colonies during periods of limited forage availability."

The introduction adequately describes the importance of honey bee nutrition and the potential benefits of dietary supplements. However, it could be improved by: Expand the background by citing relevant research on fish protein hydrolysates or similar protein-based supplements for honey bees or other insects. Clearly state the specific knowledge gap this study addresses and present the objective as a standalone statement for better clarity.

The materials and methods are explained in a straightforward manner, but a few improvements are recommended like details of Extraction Process i.e. providing more details on the FPH extraction process, including extraction time, temperature, and pH control. Clearly describe the randomization process for selecting colonies and their pre-trial health conditions to minimize bias. Specify how forager activity, brood area, and honey reserves were measured (e.g., brood measurement methods, visual observations, weight measurements). Mention which statistical tests were used to compare control and treatment groups and include confidence intervals or p-values for the reported outcomes.

The results are promising but need more quantitative and detailed information. Include numerical values for the improvements in forager activity, brood area, and honey reserves, as well as standard deviations or error bars for variability. Clearly state which outcomes were statistically significant and at what level (e.g., p < 0.05). Discuss how the results compare to previous studies on honey bee dietary supplements in more detail (if available).

Reviewer #2: Reviewer’s Notes

This manuscript presents an interesting approach to addressing nutritional deficiencies in honey bee colonies, a problem of growing concern due to its implications for both ecological balance and agricultural productivity. The authors explore the potential benefits of a tuna fish protein hydrolysate (FPH) extract as a dietary supplement for honey bees, particularly during periods of limited natural forage.

The authors hypothesized that a tuna FPH extract food supplement would boost the protein content of the supplement, leading to increased foraging activity and enhanced reserves of honey, bee bread, and brood. This is a clear and testable hypothesis.

The goals of the study as I understood them were:

Assess the efficacy of a simple and cost-effective tuna FPH extraction process.

Investigate the potential benefits of feeding a dietary supplement based on this tuna FPH extract during the winter season on colony strength and foraging activity.

These goals are well-defined and aligned with the stated hypothesis.

The major findings of the study include:

The FPH extraction process using Spasmodigestin® tablets enhanced the protein content and increased levels of approximately 15 amino acids in the tuna FPH.

Colonies fed the FPH-supplemented diet showed significant improvements, including increased forager activity, expanded brood areas, and enhanced reserves of honey and bee bread.

The study demonstrated the potential benefits of a tuna FPH-based supplement for honey bees, suggesting it could strengthen colonies and support their health during periods of limited natural foraging.

Major Comments

Line 58 = Nectar is a chemically diverse reward for bees, providing much more than carbohydrates. I think it is important to note this as you point out proteins, fats, vitamins, and minerals in pollen. Roy et al. (2017) is a good starting point, but suffice it to say that proteins, fats, vitamins, and minerals are also found in nectar and consumed by bees. These components have far reaching effects on the health of bees and the colony.

Roy, R., Schmitt, A. J., Thomas, J. B., & Carter, C. J. (2017). Review: Nectar biology: From molecules to ecosystems. Plant science : an international journal of experimental plant biology, 262, 148–164. https://doi.org/10.1016/j.plantsci.2017.04.012

Introduction

The authors state that FPH is “a very nutritious product.” Is it possible for the authors to expand upon this and describe the nutritional content, perhaps stating why it is used as a supplement in these cases (Line 100-102). This may provide a deeper conversation and comparison between these supplements and the extraction processed used here.

Suggest commenting on the importance of amino acids to honey bees and functional roles here. I know it is mentioned in the discussion, but may be important to introduce some of these comments earlier.

Line 107 = Define “nucleus colonies,” for the reader.

In the introduction, it may be beneficial to also define “colony strength.”

Line 131 = Suggest explicitly stating the modifications here, for a better understanding.

Line 146-152 = While generally understood, please provide descriptions of the chemical abbreviations (e.g. acetonitrile (ACN))

Line 164 = This is the first mention of a species, which should be described as Apis. However it may be best to insert the species in Line 50, such that it begins “Honey bees (Apis melifera or Apis spp.) play a a crucial ….”

Line 172 - Please describe the sugar used in the sugar solution. Was it glucose, fructose, sucrose, or a mixture? Was the sugar in water alone? Why was this ratio used (a citation will suffice).

Materials and Methods =

It is somewhat unclear to me how the FPH extraction method was performed and compared.I initially grasped the concept as the authors synthesized the FPH. However, in the methods and Table 1, the authors show protein, fat, and carbohydrate content of FPH before and after extraction. I then understood the comparison was between the FPH extract and canned tuna. I hope the authors can find a way to better explain this earlier in the FPH extraction section.

The rationale for using Spasmodigestin® tablets for FPH extraction could be strengthened with a more detailed explanation of why this particular product was chosen over other enzymatic methods and clarify what is meant by “well-balanced combination preparation.”

The methodology mentions that "all colonies were treated with formic acid in autumn for Varroa mites’ control..." It would be useful to know the specific concentration and application method of the formic acid, as this could potentially influence colony health and the results of the study.

While the authors explain and illustrate the remaining experiments, I would like the authors to mention or elaborate on the method of measuring consumption of the sugar solutions. It is noted that they consumed varying amounts. Yet, how were the consumption rates/quantitites measured (weight or volume)? Data in some form would be beneficial here if quantified.

It would also be helpful to understand why the 2% was chosen over the 1% and 4%. Data on the preferences for the 2% would be beneficial.

Results, Discussion, and Conclusion =

While the study shows increased forager activity, it does not delve into the potential long-term effects of this increased activity on colony health and longevity. This aspect needs further discussion if possible. The manuscript would also benefit from a deeper discussion on the observation of the significant differences between entering and exiting.

The study provides a promising foundation for future research. The use of FPH as a bee supplement is a usable approach, and the preliminary results are encouraging. However, the manuscript would benefit from a more in-depth discussion of and connections to the broader implications of the findings. For example:

How does this supplement compare to other commercially available supplements in terms of cost-effectiveness and nutritional value?

What are the potential environmental impacts of using fish by-products in this manner?

Could this supplement be adapted for use with other bee species?

Minor Comments

Spasmodigestin® is not capitalized throughout the paper, and suggest correcting this throughout.

The fonts are different for the in-line citations.

Line 1-2 = Capitalize “Novel” as well as “Applied Pilot Study,” alternatively, remove the capitalization of the other terms in the title.

Line 5 = Capitalize” University”

Line 59 = Suggest adding a “,” after “nutrition.”

Line 70 = Suggest changing the sentence to read “foraging is limited.”

Line 79 = Suggest changing the sentence to read “and easily to feed to colonies…”

Line 82 = Suggest substituting “large” or “considerable” for “big quantities.”

Line 84 = Suggest removing “totally.”

Line 86-87 = Suggest rewriting the sentence to read, “Despite this, no extensive body of research or literature uniformly supports their use.”

Line 125-126 = Italicize the species name “Katsuwonus pelamis” and add an “s” to gram to make it plural.

Line 135 = Add a hyphen to “anti-inflammatory.”

Line 145 = Suggest beginning sentence with “ Amino acid content …”

Line 149 = Suggest rewriting the sentence to read, “ FInally, 1 mL of the filtrate was dried, resuspended in 0.1 M HCl, and injected into the HPLC.

Line 173 = Add a comma after the “2” (1, 2, and 4%)

Line 177 = Suggest changing the word “quite” to “relatively.” Then if possible, provide a statement that explains what the consumption rate was slower than.

Line 180 = Suggest adding a coma after “only, as mentioned ...”

Line 181 = The authors stated the extract “smelled much better.” Great observation but could this be rephrased to a more qualitative statement? Did it smell less like fish, compared to the previous extraction?

Line 210 = Suggest removing the capitalization on “histidine.”

Line 225 = Add a hyphen to “time-dependent.”

Line 230-235 = Remove this repeated section.

Line 254-257 = Suggest rewording the statement to emphasize the consistency, “The current finding is consistent with previous research, which has shown that papain, an ingredient of Spasmodigestin® tablets, plays a significant role in protein digestion and alleviates lipid accumulation and inflammation in high-fat diets [#].” Also please update the citation to the correct format, citation 39.

Line 271 = Suggest rewording the sentence to read, “ the present study differs from the previous studies…”

Line 277 = Suggest changing “easily” to “quickly.”

Line 279 = Add the article “a” to “a typical pollen …”

Line 290-291 = Suggest rewriting sentence to read, “ Increased foraging behvaior directly results from good nutrition …”

Line 301 = Change “fats” to the singular “fat.”

Figure 2 & Figure 3 = It would be informative if the authors could include a note of statistical significance in these figures, such that readers do not have to go back and forth to the table.

6. PLOS authors have the option to publish the peer review history of their article (what does this mean? ). If published, this will include your full peer review and any attached files.

**Do you want your identity to be public for this peer review?** For information about this choice, including consent withdrawal, please see our Privacy Policy .

Reviewer #1: No

Reviewer #2: No

---

## [Author Response · Author response to Decision Letter 0]

24 Mar 2025

Response to Editor and Reviewers comments

Additional Editor Comments:

The study explores an interesting and novel approach to enhancing honey bee nutrition through a tuna-based fish protein hydrolysate (FPH) supplement, providing valuable insights into its effects on foraging activity and colony strength. However, several areas require significant revisions to improve the manuscript’s clarity and scientific depth. The introduction should provide a more detailed discussion of the nutritional components of FPH and the functional roles of amino acids in bee health. The methodology section needs further elaboration on the FPH extraction process, including the rationale for using Spasmodigestin® tablets, a clearer description of sugar syrup composition, and a quantification method for syrup consumption. The discussion should address the long-term effects of increased foraging activity on colony health and provide a comparison with existing commercial supplements. Additionally, addressing the potential environmental implications of utilizing fish by-products as a bee supplement would strengthen the manuscript. These revisions will significantly improve the manuscript’s clarity, impact, and relevance.

Response:

We appreciate the editor's positive review of the ms and the opportunity to respond to the points mentioned.

Reviewers' comments:

Reviewer's Responses to Questions

Comments to the Author

Reviewer #1:

The title is clear and informative, indicating the study's main focus and experimental outcome. However, it could be improved by making it more concise. Suggested revision:

"A Novel Fish Protein Hydrolysate Supplement Enhances Honey Bee Foraging Activity and Colony Strength: A Pilot Study.

Response:

We thank the referee for their suggestion, we edited the title as suggested.

The abstract is generally well-structured and includes the background, objective, methods, key findings, and conclusion. However, it could be improved by:

Briefly mention the duration of the feeding trial and the criteria for evaluating colony health.

Include specific quantitative outcomes to strengthen the findings (e.g., percentage increase in brood area or forager activity). Done

Use consistent terminology for "sugar syrup" (e.g., 50% W/V or 1:1 sugar:water ratio). Done

abstract revision "Honey bees require diverse nectar and pollen sources for optimal nutrition, especially during late winter and early spring. This study evaluated the effect of a tuna fish protein hydrolysate (FPH)-based dietary supplement on honey bee colonies. FPH was extracted using Spasmodigestin® tablets, increasing protein content and amino acid availability. Colonies were fed 2% FPH extract in sugar syrup weekly for six weeks, while control colonies received only sugar syrup. FPH-fed colonies showed improved foraging activity, larger brood areas, and higher reserves of honey and bee bread. These findings highlight the potential of tuna FPH-based supplements to strengthen honey bee colonies during periods of limited forage availability."

Response:

We appreciate the referee for their comment and for providing suggestions to improve the abstract, we edited it as suggested.

The introduction adequately describes the importance of honey bee nutrition and the potential benefits of dietary supplements. However, it could be improved by: Expand the background by citing relevant research on fish protein hydrolysates or similar protein-based supplements for honey bees or other insects. Clearly state the specific knowledge gap this study addresses and present the objective as a standalone statement for better clarity.

Response:

We appreciate the referee's feedback and recommendations, and we made the necessary changes. (Lines 93-96, 105-111, 124-129 ).

The materials and methods are explained in a straightforward manner, but a few improvements are recommended like details of Extraction Process i.e. providing more details on the FPH extraction process, including extraction time, temperature, and pH control. Clearly describe the randomization process for selecting colonies and their pre-trial health conditions to minimize bias. Specify how forager activity, brood area, and honey reserves were measured (e.g., brood measurement methods, visual observations, weight measurements). Mention which statistical tests were used to compare control and treatment groups and include confidence intervals or p-values for the reported outcomes.

Response:

We thank the referee for their comment. All these information is already provided in the text in section 2.3 & 2.4. We also added some details as recommended. (Lines: 136-140, 197-201)

The results are promising but need more quantitative and detailed information. Include numerical values for the improvements in forager activity, brood area, and honey reserves, as well as standard deviations or error bars for variability. Clearly state which outcomes were statistically significant and at what level (e.g., p < 0.05). Discuss how the results compare to previous studies on honey bee dietary supplements in more detail (if available).

Response:

We thank the referee for their comment. We provided numerical values as recommended (Lines 233-243). In addition, we provided all statistical outcomes in Table 3 and updated figures legends and table 3 caption. The results have been compared with previous research as recommended. (Lines 275-293).

Reviewer #2: Reviewer’s Notes

This manuscript presents an interesting approach to addressing nutritional deficiencies in honey bee colonies, a problem of growing concern due to its implications for both ecological balance and agricultural productivity. The authors explore the potential benefits of a tuna fish protein hydrolysate (FPH) extract as a dietary supplement for honey bees, particularly during periods of limited natural forage.

The authors hypothesized that a tuna FPH extract food supplement would boost the protein content of the supplement, leading to increased foraging activity and enhanced reserves of honey, bee bread, and brood. This is a clear and testable hypothesis.

The goals of the study as I understood them were:

Assess the efficacy of a simple and cost-effective tuna FPH extraction process.

Investigate the potential benefits of feeding a dietary supplement based on this tuna FPH extract during the winter season on colony strength and foraging activity.

These goals are well-defined and aligned with the stated hypothesis.

The major findings of the study include:

The FPH extraction process using Spasmodigestin® tablets enhanced the protein content and increased levels of approximately 15 amino acids in the tuna FPH.

Colonies fed the FPH-supplemented diet showed significant improvements, including increased forager activity, expanded brood areas, and enhanced reserves of honey and bee bread.

The study demonstrated the potential benefits of a tuna FPH-based supplement for honey bees, suggesting it could strengthen colonies and support their health during periods of limited natural foraging.

Response:

We thank the referee for their positive evaluation of the ms.

Major Comments

Line 58 = Nectar is a chemically diverse reward for bees, providing much more than carbohydrates. I think it is important to note this as you point out proteins, fats, vitamins, and minerals in pollen. Roy et al. (2017) is a good starting point, but suffice it to say that proteins, fats, vitamins, and minerals are also found in nectar and consumed by bees. These components have far reaching effects on the health of bees and the colony.

Roy, R., Schmitt, A. J., Thomas, J. B., & Carter, C. J. (2017). Review: Nectar biology: From molecules to ecosystems. Plant science : an international journal of experimental plant biology, 262, 148–164. https://doi.org/10.1016/j.plantsci.2017.04.012

Response:

We are grateful to the referee for pointing this up. As suggested, we included this sentence and cited this and other relevant references. (lines: 56-57)

Introduction

The authors state that FPH is “a very nutritious product.” Is it possible for the authors to expand upon this and describe the nutritional content, perhaps stating why it is used as a supplement in these cases (Line 100-102). This may provide a deeper conversation and comparison between these supplements and the extraction processed used here.

Suggest commenting on the importance of amino acids to honey bees and functional roles here. I know it is mentioned in the discussion, but may be important to introduce some of these comments earlier.

Response:

We thank the referee for their comment. As suggested, we expanded this paragraph and cited the relevant references (Lines 101-102).

Line 107 = Define “nucleus colonies,” for the reader.

Response:

As suggested, we defined it as “colonies consist of 5 combs” (Line 113)

In the introduction, it may be beneficial to also define “colony strength.”

Response:

We thank the referee for their comment. As suggested, we defined it in (Line: 129)

Line 131 = Suggest explicitly stating the modifications here, for a better understanding.

Response:

The modification here was that we kept the mixture for 12 hours at 38 C instead of 55 C for 4 hours (Line: 137).

Line 146-152 = While generally understood, please provide descriptions of the chemical abbreviations (e.g. acetonitrile (ACN))

Response:

As suggested, we defined all chemical abbreviation in (Line: 159).

Line 164 = This is the first mention of a species, which should be described as Apis. However it may be best to insert the species in Line 50, such that it begins “Honey bees (Apis melifera or Apis spp.) play a a crucial ….”

Response:

As suggested, we inserted the species name (line: 48).

Line 172 - Please describe the sugar used in the sugar solution. Was it glucose, fructose, sucrose, or a mixture? Was the sugar in water alone? Why was this ratio used (a citation will suffice).

Response:

We thank the referee for their comment. As suggested, we mentioned that the sugar type was sucrose alone with water (50% w/v) (Line 182) and we edited it throughout the ms.

Materials and Methods =

It is somewhat unclear to me how the FPH extraction method was performed and compared. I initially grasped the concept as the authors synthesized the FPH. However, in the methods and Table 1, the authors show protein, fat, and carbohydrate content of FPH before and after extraction. I then understood the comparison was between the FPH extract and canned tuna. I hope the authors can find a way to better explain this earlier in the FPH extraction section.

Response:

We thank the referee for their comment. As suggested, we clarify this in the text (Lines 138-140)

The rationale for using Spasmodigestin® tablets for FPH extraction could be strengthened with a more detailed explanation of why this particular product was chosen over other enzymatic methods and clarify what is meant by “well-balanced combination preparation.”

Response:

We thank the referee for their comment. As suggested, we clarified that this product was chosen based on the suitable composition, cost efficient and the availability in the local market in Line 147. A well-balanced combination indicates that the product is intended to enhance digestion, reduce inflammation, avoid cramps or spasms, and promote healthy bile production, all while supplying enzymes (such as papain and Sanzyme 3500) to help break down food more effectively.

The methodology mentions that "all colonies were treated with formic acid in autumn for Varroa mites’ control..." It would be useful to know the specific concentration and application method of the formic acid, as this could potentially influence colony health and the results of the study.

Response:

We used Formic acid 60% presented on carton paper. As suggested, we added this information (lines: 175-176)

While the authors explain and illustrate the remaining experiments, I would like the authors to mention or elaborate on the method of measuring consumption of the sugar solutions. It is noted that they consumed varying amounts. Yet, how were the consumption rates/quantitites measured (weight or volume)? Data in some form would be beneficial here if quantified.

Response:

We thank the referee for their comments. In these trials, we utilized normal feeders, and the consumption was assessed in volume. We discovered that at 4% FPH, bees did not consume any amount of the solution.

It would also be helpful to understand why the 2% was chosen over the 1% and 4%. Data on the preferences for the 2% would be beneficial.

Response:

We thank the referee for their comment. This concentration was chosen according to the results of the preliminary trials. As in 4% bees did not consume the solution at all. We selected 2% of FPH rather than 1% for a potential better result. We added this information (lines 194-195).

Results, Discussion, and Conclusion =

While the study shows increased forager activity, it does not delve into the potential long-term effects of this increased activity on colony health and longevity. This aspect needs further discussion if possible. The manuscript would also benefit from a deeper discussion on the observation of the significant differences between entering and exiting.

Response:

We thank the referee for their comments. As noted in the data and discussed, higher forager activity was associated with an increase in brood, bee bread, and honeycomb areas, which are commonly used to estimate colony strength. We explained this point clearly as recommended (lines: 310-313). There was no significant difference between the number of entering and exiting foragers.

The study provides a promising foundation for future research. The use of FPH as a bee supplement is a usable approach, and the preliminary results are encouraging. However, the manuscript would benefit from a more in-depth discussion of and connections to the broader implications of the findings.

For example:

How does this supplement compare to other commercially available supplements in terms of cost-effectiveness and nutritional value?

Response:

We thank the referee for their feedback. The commercial supplements are significantly more expensive than ours (around 1.6-fold). For example, one canned tuna + enzymes + extraction costs 40 Egyptian pounds (0.8 US dollars) and can feed six colonies. While a commercial pollen substitute patty costs 65 Egyptian pounds (1.3 US dollars), it only feeds four hives.

What are the potential environmental impacts of using fish by-products in this manner?

Response:

Tuna is often prepared as raw flesh and sold as loins, steaks, or canned meals. As a result, in our ms, we used commercial shredded canned tuna, and we do not employ fish byproducts like as skins, heads, bone, viscera, and muscle during loin preparation. However, waste byproducts can be processed for use in the food, feed, and pharmaceutical industries. As a result, its use provides environmental benefits such as minimizing waste and recycling nutrients, as products consider 50-70% of fish to be a good source of nutrients.

Could this supplement be adapted for use with other bee species?

Response:

We do not believe there will be any issues with this approach because it worked for honey bees and can therefore be utilized for other bee species. Future study should look into this, but at this point, more research is needed to employ this FPH with honey bees on a larger scale and over a longer period of time, as we stated in our conclusion.

Minor Comments

Spasmodigestin® is not capitalized throughout the paper, and suggest correcting this throughout.

Response: Done

The fonts are different for the in-line citations.

Response: Modified

Line 1-2 = Capitalize “Novel” as well as “Applied Pilot Study,” alternatively, remove the capitalization of the other terms in the title.

Response: Done

---

## [Decision Letter · Decision Letter 1]

8 Apr 2025

A Novel Fish Protein Hydrolysate Supplement Enhances Honey Bee Foraging Activity and Colony Strength: A Pilot Study

PONE-D-25-06788R1

Dear Dr. Al Naggar,

We’re pleased to inform you that your manuscript has been judged scientifically suitable for publication and will be formally accepted for publication once it meets all outstanding technical requirements.

Kind regards,

Muhammad Imran

Academic Editor

PLOS ONE

Additional Editor Comments (optional):

All the comments and suggestions provided by the reviewers during the initial review phase have now been thoroughly addressed and incorporated by the authors in the revised manuscript. The revised version demonstrates significant improvement and successfully meets the concerns raised. Upon re-evaluation, both reviewers have expressed their satisfaction with the revisions and have recommended the manuscript for acceptance. Therefore, based on the positive feedback and approval from the reviewers, the manuscript is now accepted for publication.

Reviewers' comments:

Reviewer's Responses to Questions

**Comments to the Author**

1. If the authors have adequately addressed your comments raised in a previous round of review and you feel that this manuscript is now acceptable for publication, you may indicate that here to bypass the “Comments to the Author” section, enter your conflict of interest statement in the “Confidential to Editor” section, and submit your "Accept" recommendation.

Reviewer #1: All comments have been addressed

Reviewer #2: All comments have been addressed

2. Is the manuscript technically sound, and do the data support the conclusions?

Reviewer #1: Yes

Reviewer #2: Yes

3. Has the statistical analysis been performed appropriately and rigorously? 

Reviewer #1: Yes

Reviewer #2: Yes

4. Have the authors made all data underlying the findings in their manuscript fully available?

Reviewer #1: Yes

Reviewer #2: Yes

5. Is the manuscript presented in an intelligible fashion and written in standard English?

Reviewer #1: Yes

Reviewer #2: Yes

6. Review Comments to the Author

Reviewer #1: The revised manuscript, titled "A Novel Fish Protein Hydrolysate Supplement Enhances Honey Bee Foraging Activity and Colony Strength: A Pilot Study," has been thoroughly reviewed, and all suggested revisions have been successfully incorporated by the authors. As all recommended changes have been adequately addressed, I now find the manuscript acceptable for publication.

Reviewer #2: I appreciate the author's attention to detail and thoroughness of their responses. The authors have adequately responded to all of my comments.

7. PLOS authors have the option to publish the peer review history of their article (what does this mean? ). If published, this will include your full peer review and any attached files.

**Do you want your identity to be public for this peer review?** For information about this choice, including consent withdrawal, please see our Privacy Policy .

Reviewer #1: **Yes: ** Muhammad Imran

Reviewer #2: No

---

## [Editor Report · Acceptance letter]

PONE-D-25-06788R1

PLOS ONE

Dear Dr. Al Naggar,

I'm pleased to inform you that your manuscript has been deemed suitable for publication in PLOS ONE. Congratulations! Your manuscript is now being handed over to our production team.

Kind regards,

on behalf of

Dr. Muhammad Imran

Academic Editor

PLOS ONE